

# Development of a web platform for the creation of automated chatbots: an innovative approach to student-teacher interaction

Carmen Lizarraga, Yadira Quiñonez, Raquel Aguayo, Jhovany Duran and Victor Reyes

Facultad de Informática Mazatlán, Universidad Autónoma de Sinaloa, Mazatlán, Sinaloa, Mexico

## ABSTRACT

This work presents an innovative solution to simplify the creation of chatbots through a web platform with an intuitive interface. The platform was evaluated based on a pilot study with 30 university students, which revealed high levels of user satisfaction with the system's usability and performance. A functional and accessible user experience is achieved using technologies like React on the frontend. The combination of Node.js and Firebase in the backend ensures efficient data management and responsive interactions. Dialogflow, a key tool in this implementation, acts as the engine behind each conversation, processing user queries and retrieving information. Furthermore, by integrating messaging APIs with Dialogflow, the project enables the deployment of chatbots in popular messaging platforms such as Telegram, WhatsApp, and Messenger, making these interactive tools easily accessible to students.

## INTRODUCTION

Today, artificial intelligence (AI) is already part of our environment and is used in various social, economic, and educational fields. Therefore, the need to incorporate AI technology into any field is growing, creating new opportunities for organizations and academic institutions to offer better products and services (*Hamad, Jamil & Belkacem, 2024*). The evolution of the digital world has witnessed constant changes and significant advances in how people and organizations interact online. With the rise of the Internet, the need for more interactive interfaces and personalized user experiences has led to the development of various technologies and approaches (*Wang, Wang & Liang, 2024*). In particular, the demand for more efficient and automated interactions on digital platforms has consolidated the design and implementation of chatbots. These virtual assistants emerged as a solution to provide rapid and consistent responses to users, eliminating barriers and improving access to information and services.

Corresponding author
Yadira Quiñonez,
yadiraqui@uas.edu.mx

Chatbots, powered by AI and natural language processing (NLP) advances, are significantly impacting the service industry. Chatbots have become increasingly popular in customer service because they can handle multiple customer queries simultaneously, provide instant personalized responses, and process orders to resolve fundamental issues without human intervention. Chatbots are typically used to automate tasks and improve user experience, so implementing them reduces operational costs, specifically in human resources, which are in charge of answering different user queries. Chatbots are now present in many applications and services on the Internet, such as Apple's assistant, Facebook's virtual assistant, and the automatic responses that many websites have, among many other examples. The most common applications of chatbots are automated customer service (*Darapaneni et al., 2022*; *Dihingia et al., 2021*), health (*Christopherjames et al., 2021*; *Softic et al., 2021*), financial transactions (*Bhuiyan et al., 2020*; *Nicoletti, 2021*), education (*Kasthuri & Balaji, 2021*; *Sophia & Jacob, 2021*), call centers (*Al-Abbasi, Elmedany & Hewahi, 2021*), and e-commerce (*Khan, 2020*; *Rakhra et al., 2021*). The applications of chatbots in the educational field are increasing; however, most chatbots are used to improve communication channels with universities, to perform automated administrative procedures, and to support personalized online counseling by solving frequently asked questions in a general way, not for a specific subject.

Currently, most universities have well-established academic advising schemes; however, they rely solely on human resources, only reaching a limited number of students, decreasing the efficiency of academic advising since a single advisor helps a small number of students. In this context, chatbots can provide significant benefits, mainly providing immediate attention and accompanying several users simultaneously in a complementary and driving way for academic advising. In this way, the chatbot can be considered a novel virtual learning technique that meets the student, physically providing an academic advising service complemented by, but not a substitute for, personal advice based on the experience of the advisors to whom the student can turn. Therefore, with the incorporation of specialized chatbots, thousands of students could receive an updated service, favoring progress toward achieving the mission of higher education.

When considering the benefits of using a chatbot in academic counseling services, it is worth mentioning that it offers several advantages for both the university or educational institution that implements it and the students who use it. For example, chatbots offer continuous service without time limits. They are accessible through different means, promoting closeness with the student and avoiding the feeling of abandonment from being unable to resolve a query at a specific time. In addition, with the dynamic, pleasant, and interactive design, they can offer standardized and personalized responses to the needs of students, reducing response time and increasing the degree of satisfaction on the part of students, which implies an improvement in the student's user experience, influencing aspects of usability and acceptance. In this sense, chatbots are a helpful tool at an institutional level because they allow for progressive customization focused on the specific information demands of students of different degree profiles.

In this context, this work proposes the implementation of specialized chatbots as a tool to provide virtual academic advice to improve the efficiency and accessibility of

educational support provided to students to resolve doubts and concerns about classes at any time, anywhere, and at any pace through instant messaging applications. To guide this research, we formulated the following questions: How can a web-based platform streamline the creation of educational chatbots? What is the impact of such a tool on student engagement and access to academic support?

Unlike general-purpose chatbot builders, this tool focuses specifically on the academic advising context, streamlining deployment in educational environments. Its main novelty is enabling teachers to create academic chatbots aligned with their curriculum content without requiring programming knowledge. Through a structured interface, teachers can input course units, topics, and resources, which are automatically transformed into conversational flows accessible to students *via* messaging applications such as Telegram. This pedagogical integration, where chatbot content is directly derived from actual course materials, constitutes a unique contribution, bridging the gap between chatbot technology and personalized academic support.

## RELATED WORK

The use of AI is revolutionizing teaching-learning in the educational environment by personalizing content according to the needs of students through the use of tools such as virtual reality, simulations, and chatbots to improve learning experiences, but above all so that students are motivated and acquire skills during the teaching-learning process and can access knowledge units at any time, anywhere, and at any pace (*Cheng, 2023*; *Zargham, 2023*). Since the launch of OpenAI's ChatGPT in November 2022 (*OpenAI, 2024*), a paradigm shift has been marked by being used by millions of users, achieving global growth, and having a positive impact on education (*Lo, 2023*; *Liu & Ma, 2023*; *Alzahrani, 2023*). In the work of *Li (2024)*, they present some application strategies of ChatGPT, including personalized learning programs, optimized intelligent language teaching, and a better understanding of intercultural communication.

The applications of chatbots in the educational field have increased considerably in recent years. The authors in *Wadhawan, Jain & Galhotra (2023)* mention that the appearance of COVID-19 had an enormous impact on education, so institutions, especially students, were strongly affected by this event, causing dropout, learning loss, and a vast digital divide. One solution was incorporating AI to support students in different institutional procedures and tasks. In *Sophia & Jacob (2021)*, the design and development of an agent for students is addressed. The agent was integrated into a website and was responsible for responding to the different intentions of students; it was developed with DialogFlow and allows integrations in Telegram, Messenger, and Slack, among others.

In another work, *Mallikarjuna Gowda et al. (2021)* propose a chatbot model that connects with the Discord platform, limited to several predetermined question answers. It works through the user's text; then, the request is sent to the server; once the information is processed, it goes to the Discord CLI, the answer to the question asked is interpreted, and finally, a response is given to the user. On the other hand, *Mendoza et al. (2022)* propose a chatbot model that acts as a guide for the student on courses taught at the institution to provide information on admission procedures, programs, and other school services,

helping with extracurricular issues, as well as in the administrative area, facilitating communication throughout the academic ecosystem. The model's objective is to have a familiar conversational environment that is simple to understand and use. The proposed model uses MySQL and Firebase storage; MySQL saves information independent of the bot, while Firebase storage is responsible for keeping information about the agent (questions and answers).

*Bilquise & Shaalan (2022)* proposes an AI-based tool to solve or deal with counseling tasks so that three AI-based alternatives are presented to address the problems that arise in the counseling process; the first is a system for recommending courses and developing study plans for subsequent semesters; the second is a machine learning algorithm that can identify students who are in danger of failing the semester in an initial period of the same; and finally, the third presents a set of chatbots that allow personalized assistance for each of the students who use it. The authors in *Bilquise, Ibrahim & Shaalan (2022)* focus on the response time for concerns that cause fatigue in the school services department. Taking advantage of AI and NLP, they developed a chatbot capable of interacting in English and Arabic; this agent was built with the Python programming language so that the architecture with which the chatbot was designed allows questions related to counseling to be asked.

*Wang, Lai & Huang (2024)* propose a specialized chatbot for programming courses; it has three main functionalities: the inquiry module, the learning module, and the Q & A module. Similarly, *Chiang, Lin & Chen (2024)* presents an educational chatbot for non-IT university students to learn programming. They use the OpenAI API to generate content, and the conversations focus on the course materials and activities. Also, *Allen, Naeem & Gill (2024)* propose a system called Q-Module-Bot that uses pre-trained large language models (LLM) to build a question-answering system that helps students with their queries and supports the delivery of education using content extracted from a virtual learning environment.

Beyond traditional academic settings, chatbots have been increasingly applied across diverse industries to enhance learning outcomes and training efficiency. For instance, conversational agents have been implemented in the automotive sector to support technical training, helping learners acquire procedural knowledge and safety protocols through simulated interactions (*Murtaza et al., 2024*, *2025*). Similarly, chatbots have been used in public policy and governmental contexts for civic education and to disseminate regulatory knowledge in a user-friendly format (*Deng & Yu, 2023*). Chatbots provide just-in-time learning, micro-training modules, and onboarding support in professional and corporate training environments, showing measurable improvements in knowledge retention and learner engagement (*De La Roca et al., 2024*). These cross-industry implementations demonstrate chatbots' adaptability and pedagogical value, reinforcing their potential to transform how knowledge is delivered and accessed, particularly when aligned with specific instructional content, as this work proposes.

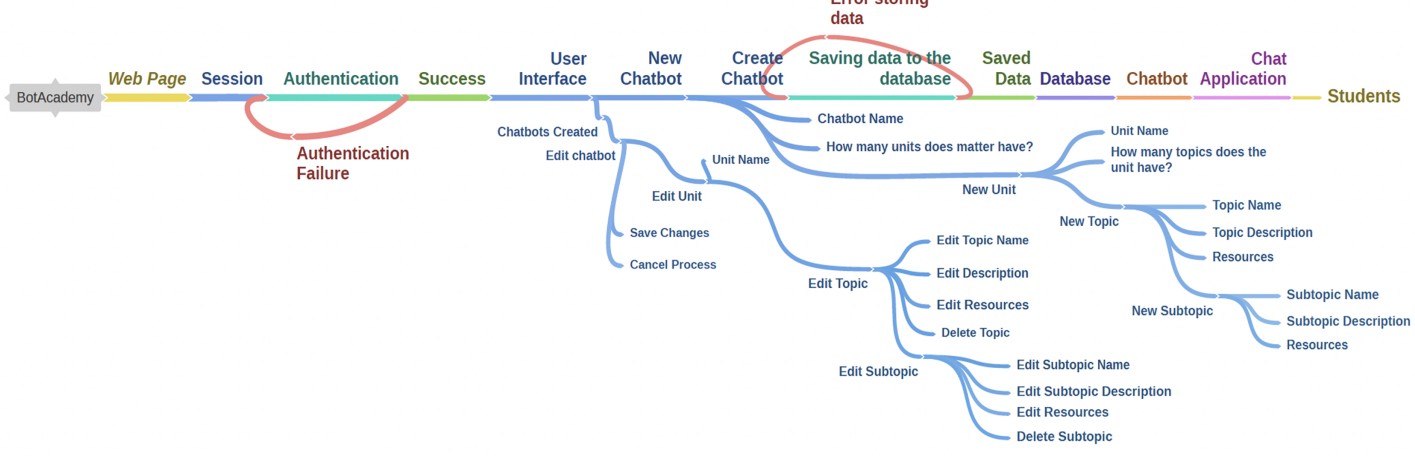

**Figure 1** General diagram of the proposal.

# DESCRIPTION OF THE PROPOSAL

The architecture proposed for the development of this work is structured with a frontend that includes the user interface intended for teachers of different subjects; in this step, the teacher must capture the number of learning units of the subject he teaches, as well as each of the subtopics that make up the study program (see Fig. 1). In addition, the platform collects all the necessary information, along with the resources, tools, and strategies required to send requests to the backend. This backend stores the information in a cloud database using Firebase services, and this collected information is vital to developing the specialized ChatBot. Finally, with this information, the chatbot is integrated into text-based platforms such as Telegram, WhatsApp, or Facebook using Firebase cloud functions. These functions connect with DialogFlow, allowing students to interact with the chatbot to obtain a response to the requested request.

Figure 2 shows the chatbot's functional architecture and describes the development tools used to achieve the interactions of the two leading actors, the teacher and the student.

# DEVELOPMENT TOOLS

## Frontend

React is a JavaScript library developed by Facebook (*Meta Platforms, Inc, 2024*), widely used for building user interfaces based on a component-driven architecture. It enables developers to create and reuse modular UI elements and integrate third-party components seamlessly. In this project, we used a syntax extension known as TSX, which allows HTML to be written within TypeScript, thereby facilitating the development of robust and type-safe interfaces. React was combined with Vite, a modern build tool that offers a significantly faster development environment (*Vite, 2024*). Node.js must be installed on the system to set up the environment as a prerequisite.

**Figure 2 Proposed architecture.**

## Backend

### Node.js

Node.js is a server-side JavaScript runtime built on Google Chrome's V8 engine (*Nodejs, 2024*). Its ability to handle multiple concurrent connections stands out, making it an ideal choice for developing scalable, high-performance applications. Node.js follows an asynchronous, event-based programming model, allowing it to efficiently handle input and output operations without blocking execution threads. This design is geared toward building network applications that handle large volumes of simultaneous traffic.

### Firebase platform

Firebase is a mobile and web application development platform developed by Google (*LLC, 2024*). It offers various cloud services that help developers build, improve, and run different applications efficiently. Offering a comprehensive set of tools for application development, this platform provides solutions such as real-time databases, authentication, and hosting, freeing developers from the burden of managing complex infrastructures.

## Dialogflow

Dialogflow is a conversational interface development platform developed by Google (*Cloud, 2024*) that is widely recognized for enabling the creation of intelligent chatbots and virtual agents. It allows developers to design conversational experiences that can understand and respond to natural language inputs, thus facilitating more natural,

efficient, and user-friendly interactions. Dialogflow supports deployment across various platforms, including mobile apps, web applications, and voice-based systems. It integrates seamlessly with multiple messaging and voice platforms such as Google Assistant, Facebook Messenger, Slack, and Telegram. Additionally, it offers multilingual support, making it suitable for building chatbots tailored to a global user base.

# EXPERIMENTAL RESULTS

This project's development is based on a Web platform that facilitates the creation of specialized chatbots through a user-friendly interface. The complete source code is publicly available at the following GitHub repository: https://github.com/JsDn4/BotAcademy, which includes frontend code (React + TypeScript), backend (Express server with Firebase integration), Dialogflow configuration, and deployment and local test instructions (README). These additions ensure that researchers or developers wishing to replicate or extend our work can do so easily.

In this context, the backend is developed using Express and Firebase Functions, exposing endpoints such as/api/login (authentication), /api/subjects (retrieving assigned subjects), and various POST routes for inserting educational content like units, topics, and subtopics into Firestore. The chatbot's logic in Dialogflow is organized using intents such as GetSubjects, GetUnits, and GetContent, with each intent trained on 10–15 example phrases and connected to specific terms related to academics. These intents are handled through a webhook that parses the user's intent from the displayName field in the JavaScript Object Notation (JSON) payload, queries Firestore using the Firebase Admin SDK, and returns context-aware responses. In addition, Dialogflow requires a dedicated webhook endpoint that manages integration with external messaging platforms such as Telegram. This integration is established through an API key provided by Telegram, which enables secure communication between the Dialogflow agent and the Telegram bot. It is important to mention that no external datasets or custom training corpora were used for chatbot training. The Dialogflow agent was configured manually using predefined intents and training phrases designed explicitly for the platform's educational context. The development team authored these training phrases with subject teachers to reflect typical student queries related to academic units, topics, and resources.

The Firestore database is organized with an explicit schema that includes collections such as teachers, subjects, units, topics, subtopics, and content, each with structured fields (*e.g.*, identifiers, descriptions, hierarchical links), ensuring coherent data management and easy mapping to chatbot responses. Regarding Firebase configuration, Firestore security rules were implemented to ensure that only authenticated users (teachers) can write or update content. At the same time, chatbot queries (read-only) are limited to public read access through a separate service account. Rules include role-based access control using custom claims on Firebase Authentication. Concerning hosting, the frontend was deployed using Firebase Hosting with HTTPS enabled and automatic SSL certificates. Continuous deployment was achieved using GitHub Actions, which triggers the build and deployment process on every push to the main branch. The backend, including all Firebase Functions (*e.g.*, /webhook, /api/units), was deployed using the Firebase CLI with Node.js runtime

```typescript
export interface Subtopic {
    id: string
    description: string
    sources: Array<Source>
    subtopicNumber: number
    subtopicTitle: string
    topicId: string
    content: Array<Content>
    relativeId: number
}

export interface SubtopicFieldProps {
    subtopicData: Subtopic
    onChange: (newSubtopic: Subtopic) => void
    onRemove: () => void
    idPrefix: number
    subtopicSources: Source[]
    onSubtopicChange: (subtopic: Subtopic) => void
    removedThings: RemovedThings
}
```

**Figure 3 Interfaces with TypeScript.**

configuration set to version 18 and a timeout of 60 s to handle Dialogflow interactions efficiently.

## Implementation of the architecture

As mentioned above, the project's frontend was developed using React with TypeScript, allowing strong typing to be implemented. This approach ensures that interfaces are clearly defined and typed, improving code consistency and robustness. The interfaces required for the project were housed in specific files, as shown in Fig. 3, providing a clear structure for managing the data and properties handled by the application's components. This use of TypeScript ensures more secure and less error-prone code.

The project follows an organized folder structure to facilitate file management. The application components are located in the components folder and are integrated into the files in the views folder, where the application views are managed. This organization is essential for the correct implementation and maintenance of the project. The helper's folder stores the auxiliary functions used to make GET and POST requests, which are necessary for the flow of information. Additionally, the Axios library is used to manage different types of requests. An example is the requestAuth() function, responsible for user authentication. These POST requests receive a username and password as parameters, are sent to the server (in this case, localhost), and wait for a response. If the request fails, it returns a server error.

```
import { initializeApp } from "firebase/app";
import { getFirestore } from "firebase/firestore";
import { getAuth } from "firebase/auth";

export const firebaseConfig = {
    apiKey: process.env.APIKEY,
    authDomain: process.env.AUTHDOMAIN,
    projectId: process.env.PROJECTID,
    storageBucket: process.env.STORAGEBUCKET,
    messagingSenderId: process.env.MESSAGINGSENDERID,
    appId: process.env.APPID,
    measurementId: process.env.MEASUREMENTID
}

const app = initializeApp(firebaseConfig)

export const db = getFirestore(app)

export const auth = getAuth(app)
```

**Figure 4 Firebase connection.**

```
app.post('/getUnits', async (req, res) => {
    try {

        const uid: string = req.body.uid
        const subjectId: string = req.body.subjectId

        const units: unitsInfo | errorType = await getUnits(uid, subjectId)

        res.send(units)
        return

    } catch (error) {

        res.send('Oops something was wrong in server.')

    }
})
```

**Figure 5 Route management with express.**

As for the backend, TypeScript was also used, albeit with a more straightforward folder structure compared to the frontend. The backend consists primarily of a folder called DB, where database-related operations are managed. At the same time, the rest of the files are organized and called from a main file called index.ts, as illustrated in Fig. 4. This simplified structure makes managing and maintaining the backend easier while keeping a clear and straightforward approach.
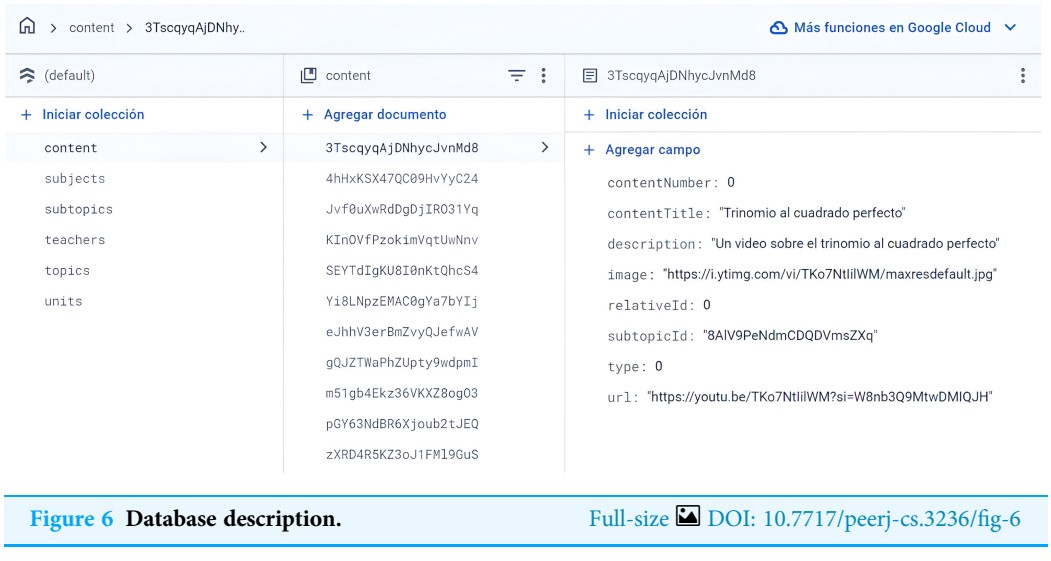

**Figure 6  Database description.**               

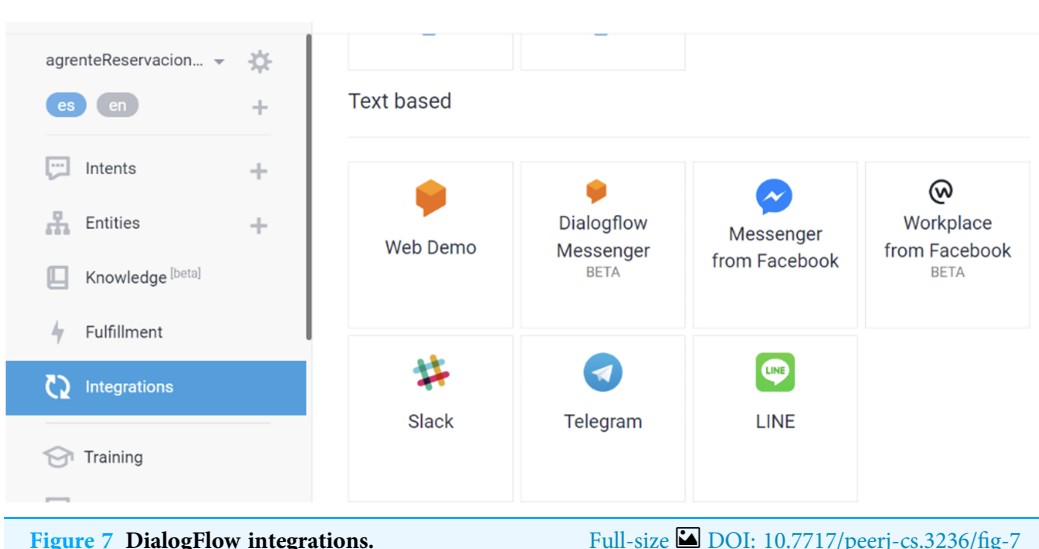

**Figure 7  DialogFlow integrations.**            

This part of the project manages the requests sent from the frontend, which update the information stored in the database and are subsequently reflected in the chatbots' responses. To do this, the Express library was used, which facilitates the creation of APIs and allows the structuring of different organized URLs. These URLs are accessible from the frontend through GET and POST requests, as shown in Fig. 5.

The database in Firebase-Firestore uses collections and documents, which requires a clear data model. The structure includes several main collections:

1. **Teachers**: This field represents teachers who use the web application and provide student resources. Its fields include subjects (an array with the IDs of the subjects they teach), teacherName (the teacher's name), and uid (the login ID).
2. **Subjects**: Contains the subjects taught. Its main field is subject (subject name).

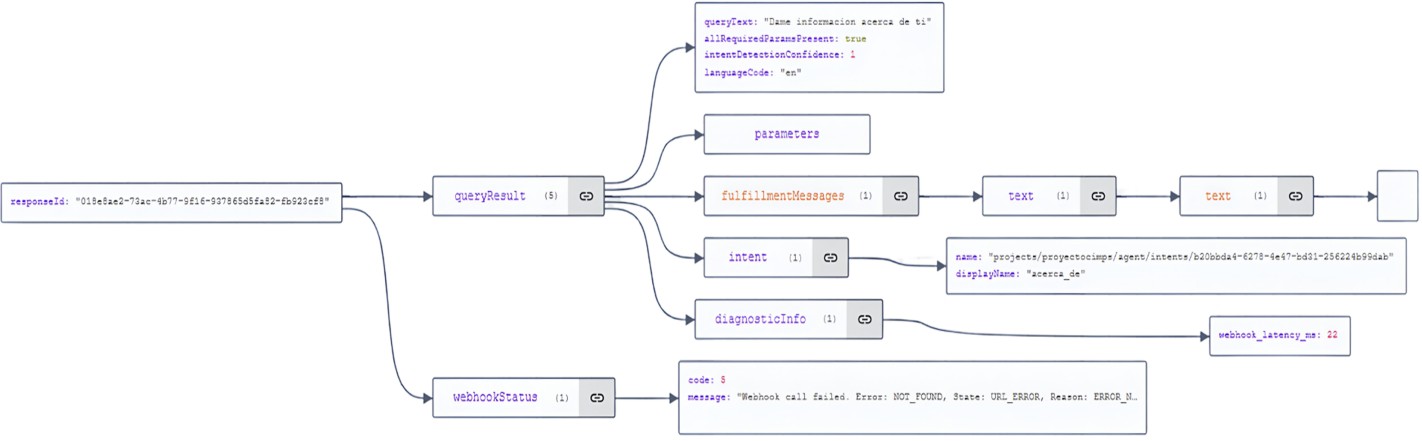

**Figure 8** JSON visualization.   

3. **Units**: This section introduces the resources for students during the course. The fields are description (unit description), relativeId (unique identifier), sources, subjectId (subject ID), teacherUid (teacher UID), unitNumber (unit number), and unitTitle (unit name).

4. **Topics**: Related to units, its fields include description (topic description), relativeId, sources, topicNumber (topic number), topicTitle (topic title), and unitId (unit ID).

5. **Subtopics**: These are subtopics within topics, with a structure similar to that of topics. They include description, relativeId, sources, subtopicTitle, and topicId.

6. **Content**: This is the final part where students access the resources. Its fields are contentNumber, contentTitle, description, image (image link), relativeId, subtopicId, type (content type), and resource URL.

Figure 6 presents the organized structure for efficiently managing the resources the DialogFlow agent shows the students.

The development of the DialogFlow flow is fundamental in the project, as it allows students to access resources provided by teachers through the created agent. To achieve this, DialogFlow was integrated with Telegram (see Fig. 7), and a cloud function linked to a DialogFlow webhook was added to manage interactions. Intents in DialogFlow identify the user's intent, determining where they are in the conversation. Using training phrases, the agent recognizes what the user wants to access. For example, for the "GetSubjects" intent, users can say words like "home" or "I want to see all subjects," and the agent will respond based on the previously defined intent.

Figure 8 illustrates how a user query is handled in DialogFlow. It shows how the query is processed, and the response is determined by evaluating intents and querying a webhook. Looking at the JSON shown in Fig. 8, it can be seen that it has initial properties that then branch out. To access the intent to be searched, the displayName property needs to be located. This property is situated in queryResult and is accessed through the intent node. Once the path to displayName is identified, its value can be accessed, allowing for a

```
 1 const functions = require("firebase-functions");          27      "text": [
 2 const express = require("express");                        28        "Response from webhook",
 3 const bodyParser = require("body-parser");                 29      ],
 4 const app = express();                                     30    },
 5 app.use(bodyParser.urlencoded({ extended: true }));        31   },
 6 app.use(bodyParser.json());                                32  ],
 7 const local = false;                                       33 };
 8 const port = 3000;                                         34 switch (displayName) {
 9 app.post("/webhook", (req, res) => {                       35   case "about":
10   const sendResponse    const { queryResult } = req.body;  36     response.fulfillmentText = "Hi, I'm a test bot";
11     const displayName = queryResult.intent.displayName;    37     response.fulfillmentMessages[1].text.text = ["Hi, I'm a test bot"];
12 const response = {                                         38     response.fulfillmentMessages.push({
13   "fulfillmentText": "Response from webhook",              39       "platform": "TELEGRAM",
14   "fulfillmentMessages": [                                 40       "text": {
15     {                                                      41         "text": ["Hi, I'm a test bot"],
16       "platform": "ACTIONS_ON_GOOGLE",                     42       },
17       "simpleResponses": {                                 43     });
18         "simpleResponses": [                               44     break;
19           {                                                45 }
20             "textToSpeech": "Response from webhook",       46 res.json(response);
21           },                                               47 if (local) {
22         ],                                                 48   app.listen(port, () => {
23       },                                                   49     console.log(`Server running on http://localhost:${port}`);
24     },                                                     50   });
25     {                                                      51 } else {
26       "text": {                                            52   exports.webhook = functions.https.onRequest(app);
                                                              53 }
```

**Figure 9  Code to display information in Dialogflow.**

comparison. With this information, the necessary programming logic can be added to send the appropriate responses to the user, thus ensuring that the interaction is fluid and relevant.

Once the user's intent has been identified within the request, it becomes essential to implement the appropriate logic to generate and return a relevant response within Dialogflow. This typically involves adding specific lines of code to configure the webhook handler, ensuring that the response aligns with the detected intent and the corresponding data retrieved from the database. Figure 9 illustrates the code implementation of a webhook using Firebase Functions, which is responsible for handling incoming requests from Dialogflow. Based on the detected intent within each request, the webhook executes the corresponding logic to retrieve relevant data and generate an appropriate response. This response is then returned to the user, enabling the virtual agent to respond dynamically to various user inputs. The architecture supports interaction across multiple platforms, including Telegram, extending the chatbot's reach and functionality.

It is important to mention that, having added a sub-path with/webhook, it is necessary to concatenate it to the initial URL, resulting in [https://initial-path/webhook]. This modified URL must be entered in the fulfillment section. When testing in the Dialogflow console, the agent responds satisfactorily to three different platforms: default response and actions on Google and Telegram, as illustrated in Fig. 10.

## Testing the interfaces

The project is structured in two main phases, aimed at facilitating and optimizing the exchange of information between teachers and students. In the first phase, teachers are

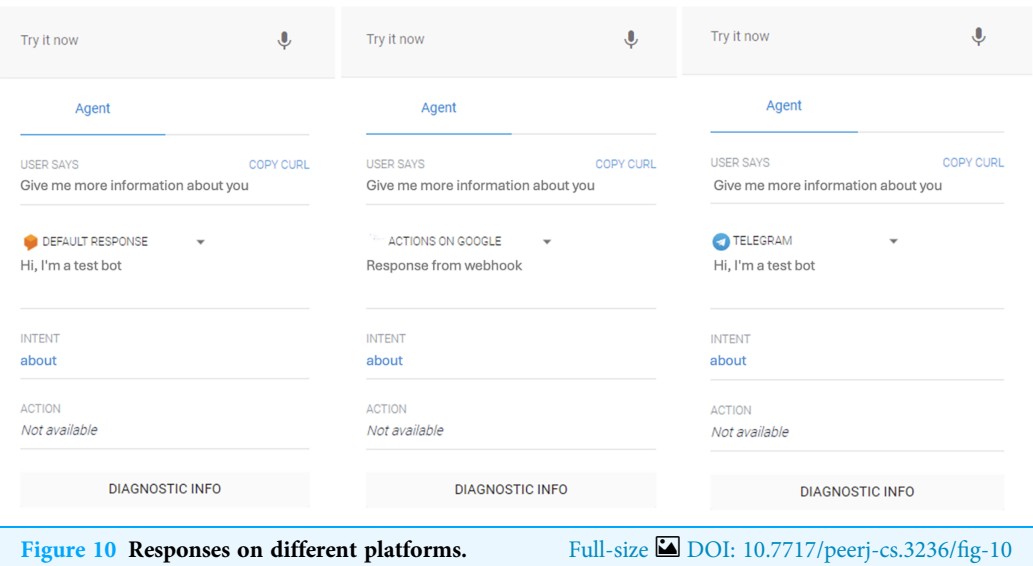

**Figure 10  Responses on different platforms.**     

responsible for updating the database with relevant information, such as educational materials and course details. The second phase is designed for students to access and use this information, allowing them to consult educational resources and important data to achieve better academic performance. Figure 11 presents a diagram describing how the user's message flows through Telegram to Dialogflow, where intent detection is performed, followed by the webhook invocation. The webhook, implemented in Firebase Functions with Express, receives the request, identifies the user's intent (*e.g.*, GetSubjects), queries the Firestore database based on intent parameters, and constructs the response returned to the user.

Regarding the interface for teachers, they must first log in to a platform designed specifically for managing educational content. After verifying their credentials through a cloud API, they can access their work area to view the subjects they teach. In this section, they can add new educational units using a form that sends the information to the database *via* a POST request once it is completed. They also have a form to modify existing data if updates are required. On the other hand, students interact with the system through a Telegram bot. To access it, they must have an account on this platform. When sending a message to the bot, it is processed by the platform and Dialogflow, which manage the database query through a POST request. If the requested data is available, Telegram returns it in JSON format. Telegram finally formats this data and presents it to the student, completing the communication and information access cycle.

### Teacher user interface

Once the teacher has logged into the developed web platform, they are presented with a workspace that displays a list of the subjects they teach. In this space, the subjects are organized and marked as pending so the teacher can enter the information corresponding to each. This allows the teacher to efficiently manage the subjects, ensuring all relevant information is entered and updated. This approach optimizes managing the subjects,

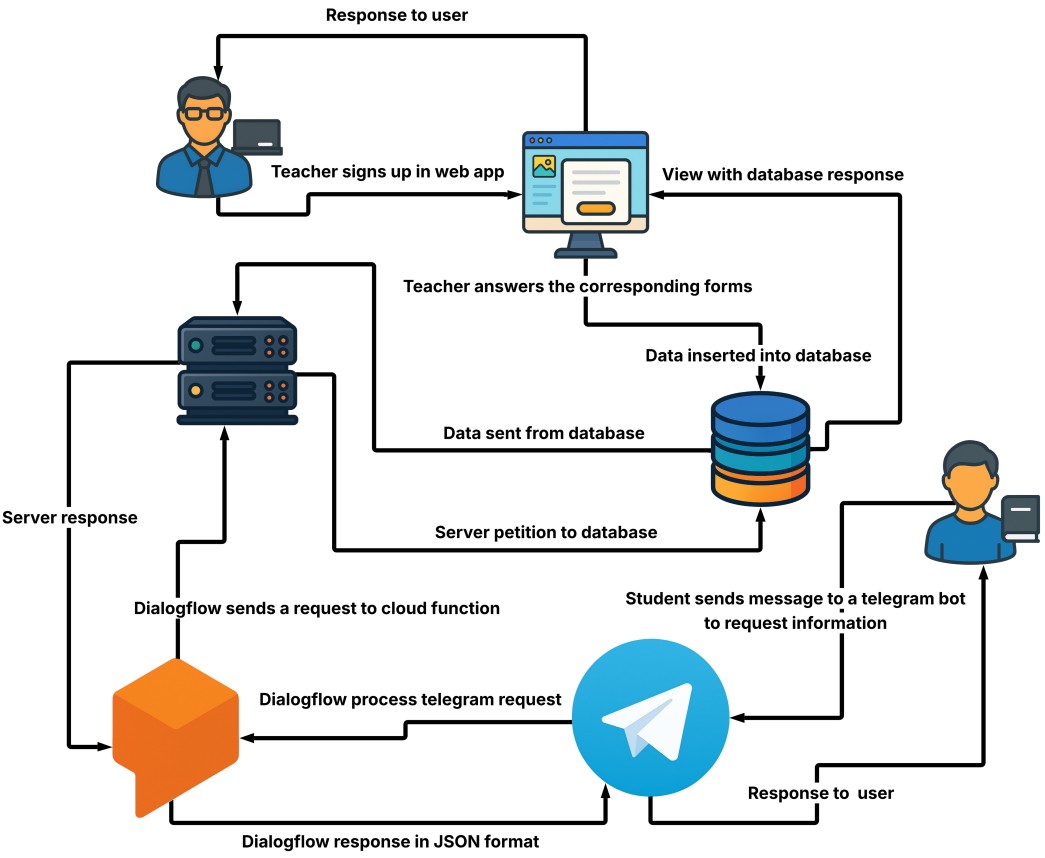

**Figure 11 Description of the chatbot development process.**

facilitating better organization and class preparation. Figure 12 illustrates this workspace, showing how the assigned subjects are presented.

The teacher will be responsible for organizing the learning units and selecting the resources that he or she considers appropriate to ensure that students can complete the assessments. To facilitate this process, a form is provided that the teacher will use to record the information and resources applied to each learning unit. This form allows the teacher to specify important details, such as the unit's objectives, the materials needed, and any additional resources that may benefit the students. Figure 13 presents a form that allows the teacher to record the information and resources used for each learning unit. This form is designed to facilitate the organization and documentation of the materials and activities carried out in the educational process.

### Student user interface

Once the teacher has completed uploading the subject information, the chatbot will be automatically generated on the Telegram platform. Figure 14 illustrates how students can interact with the chatbot to choose the unit and topic they wish to review. The interactive conversation lets students obtain specific information or advice based on their study needs.

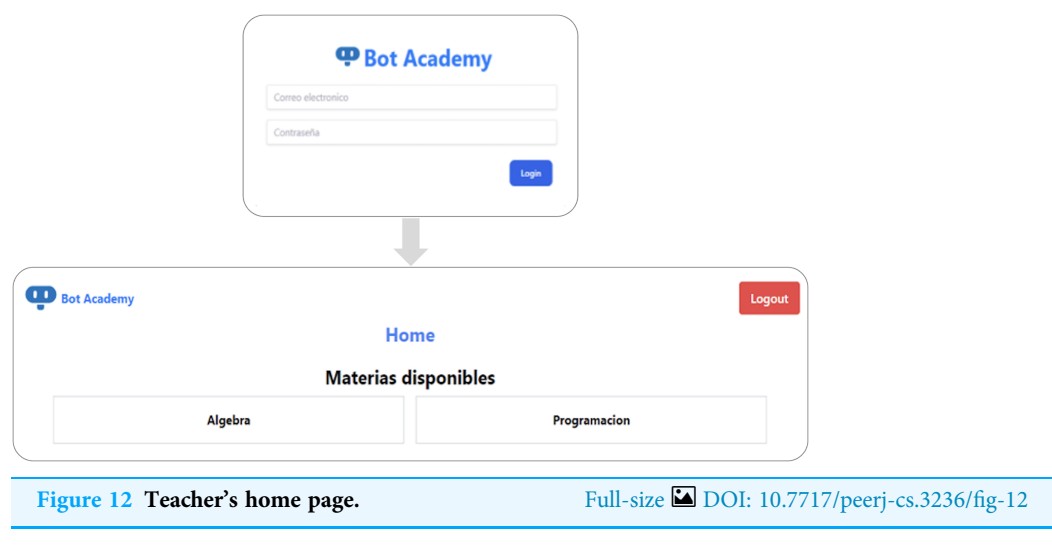

**Figure 12** **Teacher's home page.**

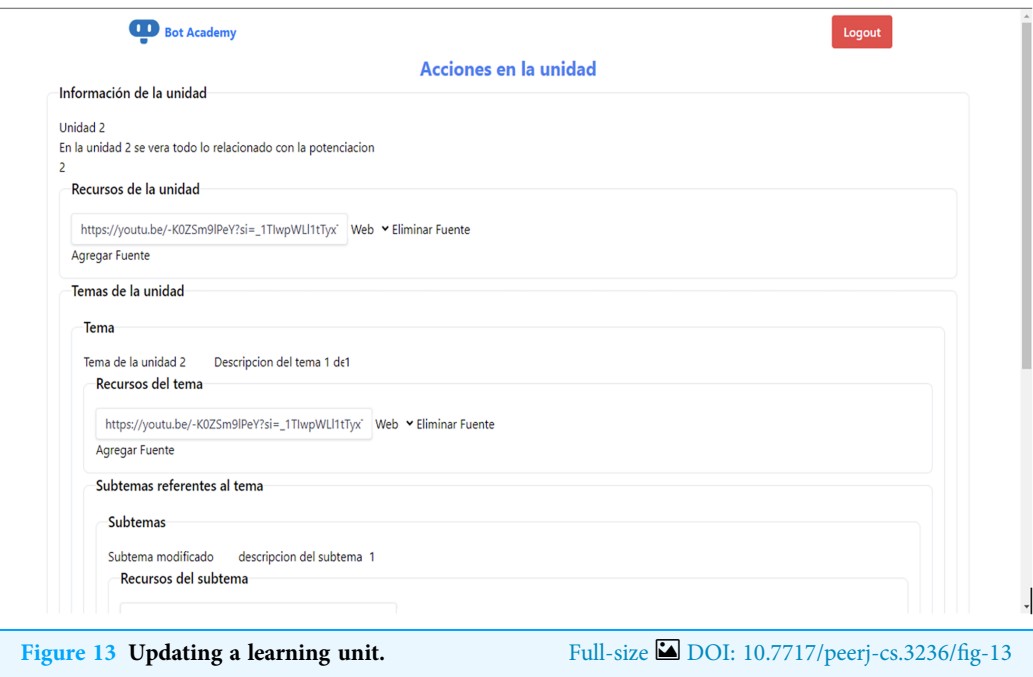

**Figure 13** **Updating a learning unit.**

The user interface is intuitive and easy to use, which enhances the experience of interacting with the chatbot.

The first time the student uses the generated chatbot (BotAcademy), the conversation must be started by clicking the "Start" button. The chatbot responds with a greeting and presents the available subject options. When selecting a subject, the corresponding options are listed. The chatbot has four subjects implemented: algebra, programming, AI, and mathematics. Figure 15 presents a simulation of how the student interacts with the chatbot implemented on the Telegram platform.

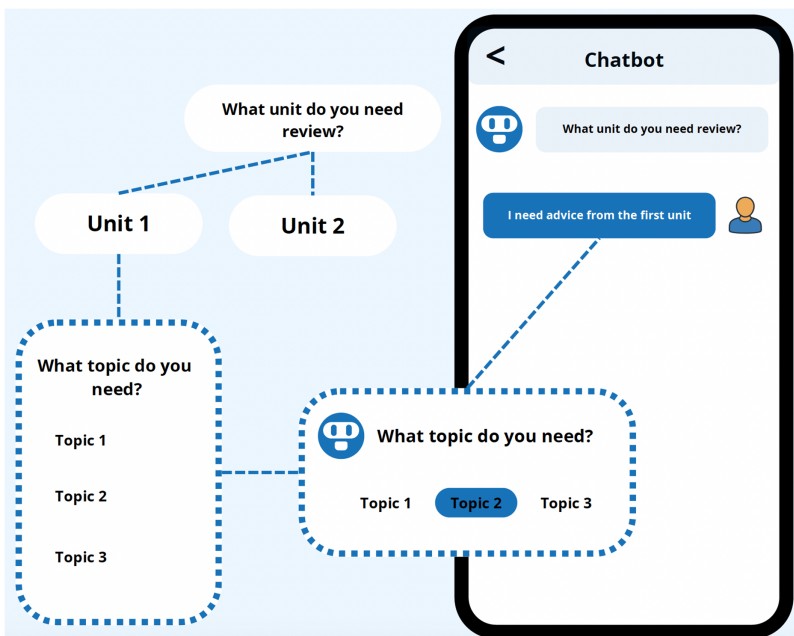

**Figure 14 Chatbot menu flow for selecting units and topics.**

**Figure 15 Simulation of the student's initial interaction with the chatbot on Telegram.**

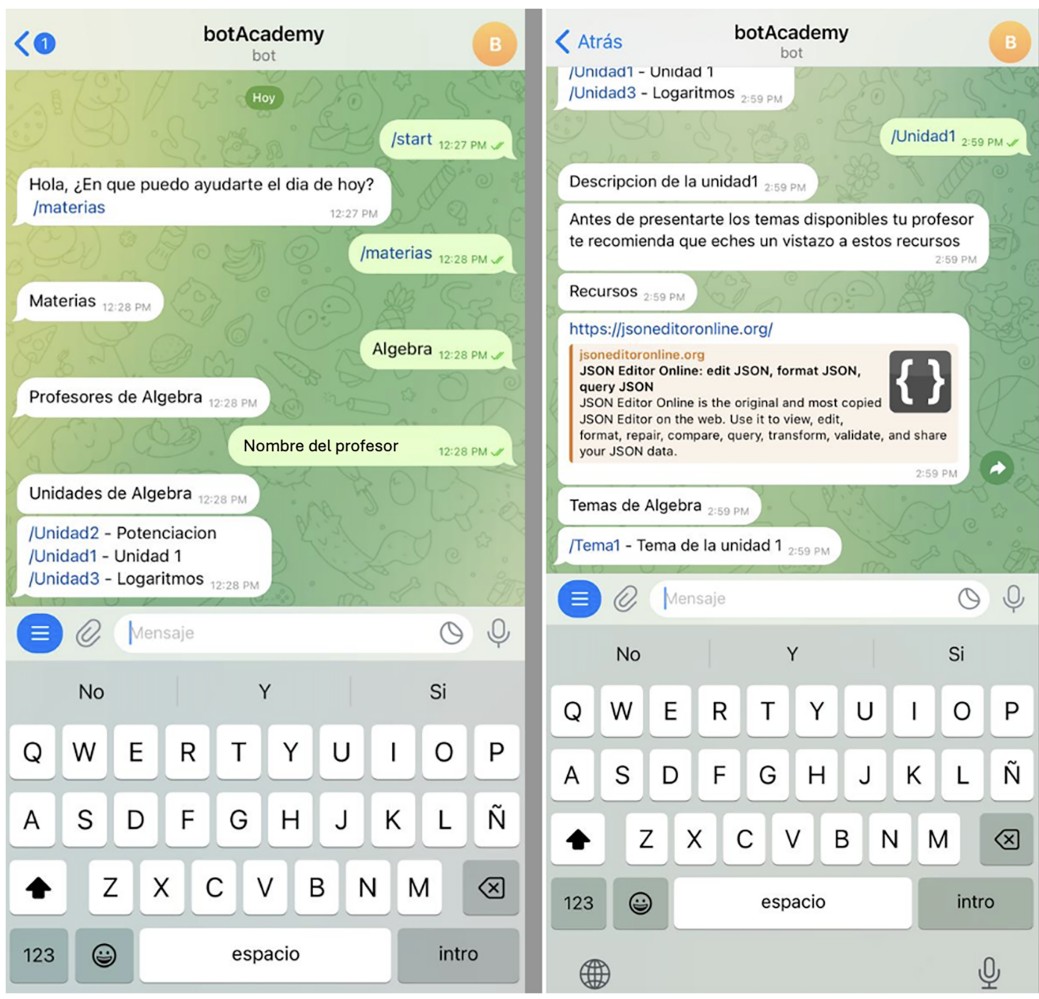

**Figure 16 Example of an interaction showing the resources for an Algebra unit.**

Several tools use NLP technologies, such as ChatGPT (from OpenAI), Google Gemini, Bard (from Google), and Bing Chat (from Microsoft), among others, that can provide students with information on specific topics. These tools allow users to obtain quick and detailed answers on various topics, facilitating access to information. However, the information they provide may be general and not necessarily aligned with the teaching methodology or resources teachers use in the classroom. In this sense, one of the main advantages of using a chatbot created by the teachers themselves is that the information it offers is based on the specific resources used by the teacher during the learning units. This allows students to have an advisory tool for various subjects, with content aligned to what has been taught in the classes of each unit. Figure 16 shows an example of the subject of algebra. As can be seen, three learning units are presented. When selecting Unit 1, information on the available topics and the resources the teacher has captured for that unit is displayed.

**Table 1 Summary of student evaluation results by category (*n* = 30).** Values represent mean scores on a 5-point Likert scale (1 = strongly disagree, 5 = strongly agree).

| Evaluation category | Mean ($\pm$ SD) |
| --- | --- |
| Ease of use | $4.6 \pm 0.5$ |
| Performance & efficiency | $4.4 \pm 0.6$ |
| Overall satisfaction | $4.3 \pm 0.7$ |
| Interactivity & features | $4.0 \pm 0.6$ |
| Interface usability | $4.7 \pm 0.4$ |
| Perception & suggestions | $3.9 \pm 0.8$ |
| Technical issues | $3.5 \pm 1.0$ |
| Support satisfaction | $4.2 \pm 0.6$ |

## Evaluation of the chatbot with students

A pilot research study was conducted with 30 undergraduates from the Faculty of Informatics at the Universidad Autónoma de Sinaloa, following the institution's ethical guidelines. Approval was granted by the academic coordination of the faculty, which determined that the protocol did not involve sensitive data and was, therefore, exempt from full ethical review. Ethical approval: Academic Coordination of the Faculty of Computer Science of the Autonomous University of Sinaloa, Approval Ref.: UAS-FIMAZ-25-034. All participants were informed about the study's objectives, and verbal informed consent was obtained before their involvement. The consent procedure emphasized the voluntary nature of participation, the anonymity of responses, and the participant's right to withdraw at any time without consequences. The students were selected through convenience sampling based on their prior experience with at least one academic advisory session. The study was carried out during a laboratory session, where students interacted with the chatbot *via* Telegram and completed a standardized evaluation instrument to assess usability, functionality, and user satisfaction.

The instrument, available at https://forms.gle/M2KEHKGvwp7M4sdv8, consists of Likert-scale items grouped into eight dimensions: ease of use, performance and efficiency, overall satisfaction, interactivity and features, interface usability, perceptions and suggestions, technical issues, and support satisfaction. Each dimension was assessed using a single-item measure rated on a 5-point Likert scale (1 = strongly disagree, 5 = strongly agree). As shown in Table 1, the descriptive statistics for each category include the mean and standard deviation.

The results suggest that most participants perceived the chatbot as intuitive, efficient, and effective. Although the feedback was positive, some students expressed the need for the chatbot to offer greater interactivity, such as access to Supplemental Materials or additional practical examples. These suggestions have been communicated to the course instructors, who are responsible for maintaining and updating the chatbot's academic resources, thus enabling iterative improvements to the student experience.

## CONCLUSIONS

This study aimed to simplify the creation of educational chatbots by providing a web-based platform that allows teachers to generate virtual agents without programming expertise. In response to the research questions regarding technical feasibility, usability, and integration with real educational environments, we demonstrated that the platform effectively enables the creation of course-specific chatbots, integrated with messaging services such as Telegram and synchronized with a real-time backend.

This work is innovative due to its twin contribution: it provides a lightweight, open-source platform for academic settings. It facilitates the automatic creation of pedagogically aligned chatbots without necessitating prior programming expertise from users. In contrast to general-purpose chatbot builders that frequently necessitate additional interfaces or are tailored for commercial applications, our platform is explicitly developed for academic advising and course assistance. The pedagogical benefit is in the teacher-led content development process, wherein educators directly organize learning units, themes, and resources, subsequently converting them into conversational flows. The solution integrates React for content entry, Firebase for real-time database operations, and Dialogflow with dynamic webhook answers, facilitating the development of responsive, topic-specific chatbots deployable across several platforms (Telegram, Messenger). This method guarantees alignment between classroom teaching and chatbot engagement, a feature commercial alternatives rarely provide. Moreover, the automatic synchronization of updated content guarantees that students continually get the latest resources, enhancing autonomous learning and maintaining continuity between teachers and students.

The pilot test results with 30 university students show high average interface usability scores ($4.7 \pm 0.4$) and ease of use ($4.6 \pm 0.5$), suggesting that the platform offered an intuitive and accessible user experience. Similarly, performance and efficiency ($4.4 \pm 0.6$) and overall satisfaction ($4.3 \pm 0.7$) indicate a positive perception of the system's functionality and utility. In contrast, lower ratings in technical issues ($3.5 \pm 1.0$) and perceptions and suggestions ($3.9 \pm 0.8$) indicate improvement opportunities, particularly in system stability and responsiveness to user expectations. Students valued the intuitive interface and speed of responses while suggesting increased interactivity and deeper content examples, as well as feedback forwarded to instructors for content improvement.

While the findings are promising, the study has some limitations. The user testing was conducted with a limited sample from a single faculty, and the chatbot's training is based on predefined phrases, which may limit scalability and flexibility. Future work should explore integrating dynamic NLP models to expand natural language comprehension, supporting multiple languages, and testing in broader academic contexts. In addition, the long-term impact of chatbot usage on learning outcomes should be evaluated through longitudinal studies.

Educational institutions must adapt their academic structures to meet the evolving needs of today's student population. In the field of educational technology, the solution

involves embracing interactive distance learning models based on two key strategies: (a) the integration of technology with course materials and (b) remote tutoring that provides motivation, feedback, and support in information retrieval and processing. When strategically designed by academic administrators, chatbots can effectively support these models, provided they are tailored to students' specific characteristics and learning needs in both traditional and distance education environments.

## ACKNOWLEDGEMENTS

The authors acknowledge the use of Grammarly, an AI-powered writing assistant, during the preparation of this manuscript. The tool was employed to support English language editing, including grammar correction, clarity enhancement, and improvement of academic writing style.

### Funding
The authors received no funding for this work.

### Competing Interests
The authors declare that they have no competing interests.

### Author Contributions
- Carmen Lizarraga conceived and designed the experiments, analyzed the data, prepared figures and/or tables, authored or reviewed drafts of the article, and approved the final draft.
- Yadira Quiñonez conceived and designed the experiments, performed the experiments, performed the computation work, prepared figures and/or tables, authored or reviewed drafts of the article, and approved the final draft.
- Raquel Aguayo conceived and designed the experiments, prepared figures and/or tables, authored or reviewed drafts of the article, and approved the final draft.
- Jhovany Duran performed the experiments, analyzed the data, performed the computation work, authored or reviewed drafts of the article, and approved the final draft.
- Victor Reyes performed the experiments, analyzed the data, authored or reviewed drafts of the article, and approved the final draft.

### Ethics
The following information was supplied relating to ethical approvals (*i.e.*, approving body and any reference numbers):

Coordinación Académica de la Facultad de Informática de la Universidad Autónoma de Sinaloa Approval Ref.: UAS-FIMAZ-25-034.

## Data Availability

Data is available at Zenodo: Lizarraga, C., & Quiñonez, Y. (2025). Evaluation Responses for Academic Advising Chatbot Pilot Study [Data set]. Zenodo. https://doi.org/10.5281/zenodo.15694426.

## Supplemental Information

Supplemental information for this article can be found online at http://dx.doi.org/10.7717/peerj-cs.3236#supplemental-information.

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
