# Peer review of "Development of a web platform for the creation of automated chatbots: an innovative approach to student-teacher interaction"

_PeerJ Computer Science, doi:10.7717/peerj-cs.3236_

## Round 0.1 · original submission · Major Revisions

Two reviewers have provided detailed feedback.

**Language Note:** The review process has identified that the English language must be improved. PeerJ can provide language editing services - please contact us at [email protected] for pricing (be sure to provide your manuscript number and title). Alternatively, you should make your own arrangements to improve the language quality and provide details in your response letter. – PeerJ Staff

Reviewer 1 ·

Basic reporting

The abstract clearly states the purpose of the work, which is to simplify chatbot creation through a web platform. The article provides sufficient field background and references.

I identified some minor errors in editing: Line 142: "stand" instead of "stands"
Lines 122-123: What did you want to say with this: "In addition, the information necessary for the development and all the resources, tools, and strategies used to send the requests to a backend."? Can this be rephrased?

Also, please rephrase the last paragraph from the conclusions, lines 318-325. It is almost unintelligible in this form.

Minor error also at Figure 2: "Intergration" instead of "Integration"
Figures 14,15,16 - their dimensions can be reduced and placed closer to their citation in text.

Experimental design

The original primary research falls under the auspices and scope of the journal.
The research questions are not present in the article. Even though the topic of the article is interesting and refers to the development of a web platform for the automatic creation of chatbots, the existence of research questions addressed within the article is required.

The structure appears well-organized, following a typical research format (Abstract, Introduction, Methods, Results, Discussion, Conclusion).

The methodology provides a logical sequence of development, describing both front-end and back-end technologies.

The results section could include more quantitative or qualitative evaluation metrics to support the claims.

Here are some areas for Improvement, to enhance replicability:
- Code or algorithmic details are missing: While technologies are mentioned, there are no specific implementation details (e.g., API endpoints, data structures, or chatbot training processes).
- Configuration details needed: The paper does not specify precise configurations (e.g., Dialogflow intents, Firebase security rules, or hosting setup).
- Experimental setup and testing: There is little to no mention of how the chatbot was tested, user interactions recorded, or success metrics evaluated.
- Dataset and preprocessing: If any training data was used for chatbot improvement, it should be explicitly mentioned and shared.

Validity of the findings

The article maintains a consistent focus on the importance of chatbot-based student-teacher interaction.

- Some transitions between sections could be smoother, especially when shifting from technical implementation to user experience discussions.

- If user testing was conducted, a clearer presentation of feedback and improvements would enhance coherence.

Additional comments

The Methods section provides a good, high-level explanation, but it lacks technical depth for full reproducibility. Including pseudo-code, workflow diagrams, or configuration specifics would help enhance clarity.

The Conclusions are relevant and generally well-stated, but they should:
- Explicitly tie back to the research question (which is not stated well)
- Include supporting results
- Acknowledge study limitations
- Provide concrete suggestions for future research.

Reviewer 2 ·

Basic reporting

-

Experimental design

-

Validity of the findings

-

Additional comments

The paper presents a timely and relevant approach to simplifying the development of educational chatbots. The proposed platform demonstrates practical integration of modern technologies such as React, Firebase, and Dialogflow, and the description of its functionality, particularly its integration with messaging platforms, is clear and informative.

However, I suggest the following improvements to strengthen the manuscript:

Abstract: Please consider including a summary of how the platform was evaluated or tested. This would significantly enhance the abstract’s credibility. Additionally, avoid vague expressions such as “pleasant experience” unless supported by specific user feedback or established UX design principles.

Novelty and Contributions: The manuscript would benefit from a clearer articulation of the platform’s novelty. What differentiates it from other existing chatbot-building platforms? Please highlight the unique technical or pedagogical contributions made by this work.

Related Work: The related work section is informative but would benefit from greater depth. Consider adding a paragraph discussing how chatbots have been applied to enhance learning outcomes across various industries, including automotive education, policy, and professional training. The following references may help support this expansion:

Figures and Visual Clarity: The included figures are relevant and helpful; however, several suffer from small or unclear text, particularly Figures 1, 2, 3, 6, and 8. Please increase the font sizes and enhance image resolution to ensure they are legible in both digital and print formats.

The system architecture is clearly described, with appropriate attention to the design and functionality of the frontend, backend, and database components. The integration with Dialogflow and messaging platforms such as Telegram, WhatsApp, and Messenger is well explained. However, the experimental design could be significantly improved in the following areas:

The research question is currently implied but not explicitly stated. The manuscript would benefit from a clearer articulation of:

The specific gap in educational chatbot tools that this work seeks to address.

The novelty and unique contributions of the proposed platform compared to existing chatbot-authoring tools.

The evaluation presented lacks methodological depth. There is no information provided regarding:

How participants were selected.

How the pilot study was administered.

What metrics were collected beyond basic user satisfaction and response accuracy?

Ethical Considerations: Since the study involves student users, it is essential to include a brief ethical statement addressing whether informed consent was obtained and whether the study adhered to institutional or national research ethics guidelines.

Initial feedback from students is encouraging, suggesting practical benefits. However, the pilot study requires more detail. Please specify the number of participants, survey design, and include any raw data or statistical analysis. Presenting this data in tables or charts would add clarity. Include more robust data and statistical analysis.

---

## Round 0.2 · Major Revisions

Reviewer 1 ·

Basic reporting

-

Experimental design

-

Validity of the findings

-

Reviewer 2 ·

Basic reporting

It looks like you have made changes. However, it is not clear whether you have addressed all the requested changes or not. As you have not provided a reference in the response letter. I would appreciate it if you could clearly mention on which page and section you have made the changes.

I can see that you have added new references regarding the use of LLM in different fields; however, some of the references are not correctly referenced. There are issues with referencing style, and the authors' details are wrong.

For example title is correct, however author details are wrong.
The impact of LLM chatbots on learning outcomes in advanced driver assistance systems education

Murtaza, M., Cheng, C.T., Albahlal, B.M., Muslam, M.M.A. and Raza, M.S., 2025. The impact of LLM chatbots on learning outcomes in advanced driver assistance systems education. Scientific Reports, 15(1), p.7260.

Murtaza, M., Cheng, C.T., Fard, M. and Zeleznikow, J., 2024. Transforming driver education: a comparative analysis of LLM-augmented training and conventional instruction for autonomous vehicle technologies. International Journal of Artificial Intelligence in Education, pp.1-38.

Experimental design

-

Validity of the findings

-

---

## Round 0.3 · accepted · Accept

After two rounds of review, I am pleased to inform you that your paper has been accepted for publication. The reviewers are satisfied with the revisions, and no further major concerns remain.

Please ensure that the final version includes the Ethical Committee approval reference number in the “Evaluation of the Chatbot with the Students” section, as requested by one reviewer.

Congratulations on this achievement, and thank you for your contribution.

Reviewer 2 ·

Basic reporting

Thank you for making the changes. I am fine with the current version of the paper.

However, in the final version, please address the following comment.

In the "Evaluation of the Chatbot with the students" section, please add the Ethical Committee approval reference number of the study.

Experimental design

-

Validity of the findings

-